# Polymer Ligands Derived from Jute Fiber for Heavy Metal Removal from Electroplating Wastewater

**DOI:** 10.3390/polym12112521

**Published:** 2020-10-29

**Authors:** Md Lutfor Rahman, Choong Jian Fui, Tang Xin Ting, Mohd Sani Sarjadi, Sazmal E. Arshad, Baba Musta

**Affiliations:** Faculty of Science and Nature Resources, Universiti Malaysia Sabah, Kota Kinabalu 88400, Malaysia; jianfui@hotmail.com (C.J.F.); tangxinting_0917@hotmail.com (T.X.T.); msani@ums.edu.my (M.S.S.); sazmal@ums.edu.my (S.E.A.); babamus@ums.edu.my (B.M.)

**Keywords:** adsorption, jute cellulose, graft copolymer, poly(hydroxamic acid), poly(amidoxime) ligand, wastewater

## Abstract

Industrial operations, domestic and agricultural activities worldwide have had major problems with various contaminants caused by environmental pollution. Heavy metal pollution in wastewater also a prominent issue; therefore, a well built and economical treatment technology is demanded for pollution-free wastewater. The present work emphasized pure cellulose extracted from jute fiber and further modification was performed by a free radical grafting reaction, which resulted in poly(methyl acrylate) (PMA)-grafted cellulose and poly(acrylonitrile)-grafted cellulose. Subsequently, poly(hydroxamic acid) and poly(amidoxime) ligands were prepared from the PMA-grafted cellulose and PAN-grafted cellulose, respectively. An adsorption study was performed using the desired ligands with heavy metals such as copper, cobalt, chromium and nickel ions. The binding capacity (q_e_) with copper ions for poly(hydroxamic acid) is 352 mg g^−1^ whereas q_e_ for poly(amidoxime) ligand it was exhibited as 310 mg g^−1^. Other metal ions (chromium, cobalt and nickel) show significance binding properties at pH 6. The Langmuir and Freundlich isotherm study was also performed. The Freundlich isotherm model showed good correlation coefficients for all metal ions, indicating that multiple-layers adsorption was occurred by the polymer ligands. The reusability was evaluated and the adsorbents can be reused for 7 cycles without significant loss of removal performance. Both ligands showed outstanding metals removal capacity from the industrial wastewater as such 98% of copper can be removed from electroplating wastewater and other metals (cobalt, chromium, nickel and lead) can also be removed up to 90%.

## 1. Introduction

Water is an essential element for living organisms, including humans, animals especially aquatic organisms, as well as plants. Therefore, water is vital to the environment for the persistence of life on Earth. However, owing to flash urbanization, this has led to myriads of human activities such as high industrialization and agricultural activities. Consequently, water quality is highly affected and deteriorated. Water pollution, therefore, has become one of the greatest issues and concerns in the world [1]. Besides, heavy metals present in the polluted water, especially wastewater are one of the biggest causes of water contamination issue. The presence of low concentrations of these heavy metals is crucial to ensure biochemical and physiological activities in living organisms function well. Heavy metals can caused lethal effects, when they come in high concentrations [2]. These non-biodegradable heavy metals are generally created and present as factory sewage and waste from many factory processes such as electroplating, electrolysis depositions, milling, and metal coatings. A high concentration of these hazardous heavy metals is responsible for many detrimental health effects such as dermatitis, mental retardation, weakened immune system, liver damage and dysfunction, alteration of genetic materials, and kidney failure, which are often reviewed by known international bodies such as the World Health Organization (WHO) and that can last for a very long period of time [3,4].

Several conventional methods or treatments for removing these heavy metals can be accomplished. For instance, adsorption [5], precipitation [6], coagulation [7], reverse osmosis [8], and membrane filtration [9]. Among these conventional methods, adsorption has been proven to be a very promising and cost-effective method and basically, adsorbent materials are made from low-cost waste or biomass known as bio-adsorbents [10]. Cellulosic materials are among the good candidates for use and are modified to become bio-adsorbents. However, cellulose itself has low sorption capacities. Therefore, chemical modification is required to enhance its effectiveness on sorption capability using grafted cellulose converted into the chelating ligands as an efficient adsorbent [11]. 

The most conventional method to carry out graft copolymerization is by the free radical polymerization method. There are several reasons that radical polymerization is widely used. This includes its ability to produce unlimited number of copolymers, good water tolerance and other impurities, involves simple steps, and is relatively cheap compared with other competitive and sophisticated technologies. It is a reaction involving the presence of radicals, and it is a chain reaction process, consisting initiation, propagation and termination [12]. An initiator is required to initiate the chain reaction. For graft copolymerization of cellulose, free radicals can be formed by initiators such as ceric ammonium nitrate (CAN) as an oxidizing agent, or via irradiation with gamma rays or ultraviolet rays. Radicals generated on the cellulose will form covalent bonds with the monomers (methyl acrylate, acrylonitrile, styrene etc.) introduced, resulting in another newly formed free radical site. As a result, more and more monomers can be subsequently added to the free radical site of the terminal, propagating extensively until the termination process occurs, in which the growing chain is stopped from growing longer, via the disproportionation termination step or combination of two growing cellulose chains as the termination process [13,14,15,16].

Subsequently, chemical modification of the grafted cellulose using various reactions offer suitable chelating ligands for its promising enhancement on the coordination with metals ions. In general, a coordination compound is actually composed of atoms, molecules or small groups of atoms that contain electrical charges which are usually negatively charged or neutral charge, that are attached to a central metal atom [17]. Metal complexes or central metal ions generally function as a Lewis acid as they are positively charged. On the other hand, the ligand functions as a Lewis base as it carries lone pairs of electrons that will actually share with the attached central metal ion, resulting in the formation of a coordinate covalent bond. Coordination compounds are essential and vital role in many chemistry related fields such as analytical chemistry, inorganic chemistry, biological systems, industry, and medicine [18].

In this study, pure cellulose is extracted from jute fiber. Then the cellulose was used in a grafting reaction with methyl acrylate/acrylonitrile monomers by the free radical initiation method to yield poly(methyl acrylate)-grafted cellulose (PMA-grafted cellulose) and/or poly(acrylonitrile)-grafted cellulose (PAN-grafted cellulose). Both grafted celluloses were converted into the poly(hydroxamic acid)/poly(amidoxime) chelating ligands, which is used in the removal of heavy metal ions from synthetics and wastewater. The adsorption ability of heavy metals is relatively higher as they are chelates with multiple bonds that uptake metal ions in a short period of time and, most importantly, it is reusable in several cycles. 

## 2. Materials and Methods

### 2.1. Chemicals

Jute known as golden fiber (Tossa jute) was obtained from the Machpara, Pansgha, Bangladesh. Other chemicals were purchase from local market and list of chemicals are provided in the Appendix A. 

### 2.2. Extraction of Cellulose from Jute Fiber

For removing the lignin component, about 250 g of dry jute fiber was boiled with 1 L of 15% sodium hydroxide (NaOH) solution and bleach with hydrogen peroxide (50%) until the mixture become white color. Details parameters and procedure are described in the Appendix A. 

### 2.3. Preparation of Poly(Methyl Acrylate)/Acrylonitrile (PMA/PAN)-Grafted Jute Cellulose

The PMA-grafted cellulose/or PAN-grafted cellulose was synthesized from the jute cellulose using free radical initial process. The methyl acrylate monomer was used for the synthesis of PMA-grafted cellulose whereas PAN-grafted cellulose was prepared from acrylonitrile monomer. For graft copolymerization, reaction conditions and parameters are described in the Appendix A [11,15]. 

### 2.4. Synthesis of Poly(Hydroxamic Acid) Ligand or Poly(Amidoxime) Ligand

The poly(hydroxamic acid) ligand was prepared from PMA-grafted cellulose whereas poly(amidoxime) ligand was prepared from PAN-grafted cellulose using an oxidation reaction. For the preparation of hydroxylamine solution and polymer ligands, the reaction conditions are described in the Appendix A [11,15]. 

### 2.5. Adsorption and Isotherm Study by Poly(Hydroxamic Acid)/Poly(Amidoxime) (PHA/PA) Ligands 

The batch adsorption was utilized to study the adsorption characters of the synthesized polymer ligands (poly(hydroxamic acid)/poly(amidoxime), PHA/PA) with four types of metal ions selected at different pH from 3–6. The lower and higher concentration of metal ions solution (10 to 1800 ppm) was used for isotherm study. Detailed batch adsorption and isotherm study conditions are described in the Appendix A. The inductively coupled plasma - optical emission spectrometry (ICP-OES) was used to measure the initial and final concentration of the metals ions by Equation (1) [11,15,19].
(1)$$q_{e}=\frac{\left(C_{o}-C_{e}\right)V}{m}$$
where q_e_ stand for adsorption capacity (mg g^−1^). Here C_o_ and C_e_ are the initial and the final concentration of the metal ions (mg L^−1^) in solution, respectively. V and m are referred to the volume (L) and the mass of the adsorbent (g), respectively. 

### 2.6. Kinetic Study

A kinetic adsorption study was carried out using the ligands (PHA/PA) with selected metal ions solution and detail kinetic adsorption conditions are described in the Appendix A. The final metal ions concentration was measured by using the ICP-OES and the metals ions were measured at particular times by Equation (2) [10,15,19].
(2)$$q_{t}=\frac{\left(C_{o}-C_{t}\right)V}{m}$$
where q_t_ stand for adsorption capacity at time, t (mg g^−1^). C_o_ and C_t_ are the initial and the final concentration of metals ions (mg L^−1^), respectively. V and m refer to the volume (L) and the mass of the adsorbent (g), respectively.

## 3. Results and Discussion

### 3.1. Reaction Mechanism

The free-radical reactions have been extensively used for the creation of free radical on the starch or cellulose structure and subsequently reactions with vinyl or acrylic monomers [14,20]. One of the alternative processes among several other methods, transition metal ions are created by the free radicals on the O atom of primary OH groups of the β(1→4) linked D-glucose units [19]. The free radicals bearing D-glucose units are then added with acrylic monomers and further initiated the free radical on the cellulosic backbone for the propagation reaction resulting in the grafting copolymer products. The current study emphasises the graft copolymerization of jute cellulose with methyl acrylate/or acrylonitrile initiated with the ceric ions as a free-radical chain reaction (Scheme 1). The ammonium cerium(IV) nitrate was used as a source of ceric (IV) ions, which can be created a complex of the D-glucose units, and the hydrogen atom is oxidized by the reduction of the Ce^4+^ to Ce^3+^ ions. Thus, the free-radicals on the cellulose cause the initiation of the double bonds in the monomers, and the subsequent propagation and termination reactions have been accomplished expected by the combination of grafting, as shown in Scheme 1 [16].

In this study, two types of monomers as such the methyl acrylate (MA) and acrylonitrile (AN) were used separately for jute cellulose grafting process according to the methods reported in our earlier papers [14,21]. The graft copolymerization reaction involving the jute cellulose, monomers, initiator (ceric ammonium nitrate, CAN) with optimal conditions were used according to our earlier paper [13,21]. The PMA-grafted cellulose/or PAN-grafted cellulose copolymers were then converted to chelating ligands such as the poly(hydroxamic acid) ligand (PHA)/or poly(amidoxime) ligand (PA), respectively. Both the chelating ligands were synthesized from their respective grafted copolymers, using hydroxylamine hydrochloride and methanolic solution. The chemical conversion was carried out under reported optimal conditions such as pH 10 and temperature is 70 °C for 4 h [20,22]. As expected, the appearance of PHA chelating ligand is white in color, while PA chelating ligand is yellowish brown in color according to previous studies [23,24]. These PHA/PA are used as the adsorbents for heavy metal extraction in the batch adsorption process. During adsorption process, PHA chelating ligand adsorbed Cu^2+^ ions to form PHA–Cu complex, while PA chelating ligand adsorbed Cu^2+^ ions, forming a PA–Cu complex. Different metal ligand complexes were formed depending on the type of heavy metal ions being adsorbed by the chelating ligands as shown in Scheme 1.

### 3.2. Fourier-Transform Infrared Spectroscopy (FT-IR) Spectroscopy Analysis

Figure 1a shows the FT-IR spectra of jute cellulose and it can be clearly seen that a small peak which is observed at 886 cm^−1^, indicating the presence of C_1_–H deformation of glycosidic linkage between glucose units in the cellulose structure [25]. Relatively a broad infrared (IR) band shows at 1052 cm^−1^, which is indicated the presence of the vibration of C–O–C pyranose ring of a cellulosic structure. Besides, a small peak is visible at 1158 cm^−1^ is assigned to the stretching of CO and 1427 cm^−1^ representing a deformation vibration for CH_2_ in the anhydroglucose units (AGU) structure, in addition a bending mode also observed at 1640 cm^−1^ belongs to the bound water. The peak detected at 2897 cm^−1^ representing the stretching of C-H, while the peak seen at 3333 cm^−1^ indicated the O–H stretching bond in the structure of cellulose (Figure 1a). 

The IR spectrum of poly(methyl acrylate)-grafted cellulose (PMA-grafted cellulose) is shown in Figure 1b. It is clearly found that there was a new significant sharp peak at 1727 cm^−1^, which represent the stretching band of carbonyl C=O groups, belongs to the methyl acrylate. This significance band appeared to be the graft copolymerization reaction in which the monomer methyl acrylate was used in the process of grafting on to the jute cellulose. Another two significant sharp peaks at 1450 and 827 cm^−1^ were found for C–H bending scissoring for CH_2_ and wagging for CH_2_, respectively [26]. Despite the new peak found in the spectrum of PMA-grafted cellulose, other peaks were retained, which is belongs to the cellulose structure [27]. 

The spectrum of the poly(hydroxamic acid) (PHA) ligand is shown in Figure 1c. A broadened peak was found at 3119 cm^−1^, which indicated the presence of N–H stretching. A peak formed at 1397 cm^−1^ caused the O–H bending mode for N–O–H. Indeed, the significant sharp peak found in the spectrum of PMA-grafted cellulose at 1727 cm^−1^ was vanished and there were two bands formed at 1677 cm^−1^ and 1648 cm^−1^ in the hydroxamic acid, certainly designated to the stretching of C=O of the amide group [26]. In addition, the bands appeared at 2947 cm^−1^ and 2866 cm^−1^ to the C–H stretching of methylene (CH_2_) and methyl (CH_3_) groups, respectively. The noteworthy data about the IR spectrum of PHA shows a band in the region around 2750 cm^−1^ for O–H stretching, which is typically hydroxamic acid functional group. This O–H stretching is characterized by a greater magnitude due to the hydrogen bridge forms between the oxygen and hydrogen atoms [26]. This IR spectra (Figure 1c) proved that the successful conversion of PMA-grafted cellulose into poly(hydroxamic acid) ligand [27].

In Figure 1d, the spectrum representing poly(hydroxamic acid)-copper complex, it can be noticed that N–H stretching at 3119 cm^−1^ and O-H stretching at cm^−1^ were significantly affected due to the formation of metal-ligand complex. Moreover, the peak of the poly(hydroxamic acid) at 1677 cm^−1^ (C=O stretch) is totally vanished due to C=O form a complex by coordination bond with copper ions and another C=O amide is retained at 1652 cm^−1^. This confirmed the formation of metal-ligand complex in which the copper ions were adsorbed by the PHA ligand [28].

In Figure 2a, IR spectra obtained from the jute cellulose are similar with Figure 1a. The second IR spectrum for the poly(acrylonitrile)-grafted cellulose (PAN-grafted cellulose) is represented by the Figure 2b. A typical nitrile group, C≡N stretching band can be clearly observed that a new noteworthy sharp peak at 2242 cm^−1^ due to the presence of nitrile groups belonging to the acrylonitrile functional groups. The graft copolymerization reaction was successfully occurred on the monomer acrylonitrile onto the jute cellulose. Despite having the new peak found in the spectrum of PAN-grafted cellulose, other peaks were retained, which belong to the structure of jute cellulose [27].

In Figure 2c, a spectrum represents the poly(amidoxime) ligand, it can be observed that there was a broadened peak found at 3285 cm^−1^, due to the presence of N–H stretching. Indeed, a peak was seen at 1425 cm^−1^ due to the O–H bending mode. It can be noticed that the significant sharp peak found in the spectrum of PAN-grafted cellulose at 2242 cm^−1^ was vanished and a new absorption band at 1681 cm^−1^ belongs to the C=N stretching mode, indicated the presence of amidoxime functional groups in the ligand (Figure 2c).

### 3.3. Field Emission Scanning Electron Microscopy (FE-SEM) Analysis

Some morphological analyses were carried using the FE-SEM technique with JSM-7800F (JEOL) and JSM-7900F (JEOL). The SEM micrograph of jute cellulose appeared the waviness and folding surface with sticky morphologies (Figure 3a) using JSM-7800F. Other SEM micrographs were obtained by JSM-7900F and such SEM of the PMA-grafted jute cellulose appeared with a non-smooth surface where the waviness and folding surface is abolished resulting in the noticeable grafting event on the cellulosic structure as shown in Figure 3b. No homopolymer is detectable on the surface of PMA-grafted cellulose. In Figure 3c, the SEM image of poly(hydroxamic acid) ligand showed completely different morphology compared to the grafting cellulose. The partially smooth surface of the PMA grafted cellulose was cracked and formed uneven spherical beads with differing sizes as shown in Figure 3c. The spherical beads were abolished forming compact morphologies when poly(hydroxamic acid) ligand adsorbed the copper ions as characterized by Figure 3d.

The same jute cellulose (SEM micrograph at 4a is similar with 3a) was used to prepare poly(acrylonitrile) grafted jute cellulose. The SEM micrograph of the PAN-grafted cellulose showed unsmooth and rough surface morphologies, which were distinguishable with pure jute cellulose (Figure 4b). No homopolymer was detectable on the surface of PAN-grafted cellulose. The SEM of poly(amidoxime) ligand appeared well-defined morphologies with larger size of spherical beads as shown in Figure 4c. In addition, the poly(amidoxime) ligand displayed small sized spherical beads after bind with copper ions due to shrinking of ligand surfaces as shown in Figure 4d.

### 3.4. X-ray Diffraction (XRD) Analysis

The X-ray diffraction (XRD) analyses were performed with Rigaku automated multipurpose XRD using CuKα at 40 KV and 50 mA. Scattered radiation was detected in the range of 2θ = 3–80°, at a scan rate of 4°/min. Cellulose, is one of the most abundant polymers that can be found naturally, having both crystalline and amorphous phase [29]. The XRD pattern of cellulose, cellulose-graft copolymers, and its respective chelating ligands obtained were compared with XRD patterns obtained by other reported works [29].

Figure 5a showed the XRD patterns from the jute cellulose which was then chemically converted into cellulose-supported poly(hydroxamic acid) ligand. Based on the Figure 5a, it can be noted that there were three peaks were found in the XRD pattern of cellulose such as 2θ = 15.5°, 22.6°, and 34.5°, which were matched with the XRD patterns of raw cellulose as untreated fibre and reported study [29]. There are two well-defined sharp peaks present in the cellulose, one at 2θ = 15.5° indicating the crystalline phase of a cellulose, and another peak at θ = 22.6°, showing the non-crystalline or amorphous phase of a cellulose [30].

In Figure 5b, four peaks are found at 2θ = 7.0°, 11.0°, 12.8° and 17.0° indicating the crystalline areas of cellulose-graft copolymer due to the presence of PMA, while peak at 2θ = 22.6° was retained in the XRD pattern of cellulose as the amorphous nature of the jute cellulose-graft-PMA [31]. In Figure 5c, the XRD pattern of the synthesized PHA ligand peaks belongs to PMA at 2θ = 7.0°, 11.0° and 12.8° was disappeared and new peaks appeared at 2θ = 30.0° and 32.0°. This confirmed the successful chemical conversion of methyl acrylate functional groups into hydroxamic acid functional groups (PHA chelating ligand), while the cellulose peak originally found at 2θ = 15.5° and 22.6° were retained for PMA-grafted cellulose. Figure 5d show the XRD pattern of the PHA ligand after complexation with Cu(II) ions, forming a PHA-Cu complex. Three new peaks were found in Bragg’s reflections at 2θ = 43.3°, 50.4° and 74.1°, which can be indexed as (111), (200), and (220) copper planes, confirming the formation of PHA–Cu complex [32].

On the other hand, Figure 6 showed the XRD patterns from the jute cellulose which was chemically converted into cellulose-supported poly(amidoxime) ligand.

The XRD pattern obtained after acrylonitrile grafted cellulose (PAN-grafted cellulose) is shown in Figure 6b. As a result, there was a sharp peak at around 2θ = 17.1°, indicating the ordered structure of polyacrylonitrile (PAN) in the PAN-grafted cellulose. It can be noted that the certain peak intensity of PAN-grafted cellulose was smaller than that in cellulose (2θ = 22.6°) due to the presence of PAN particles which affected the crystalline phase of cellulose [33]. Figure 6c shows the XRD pattern of cellulose-supported polyamidoxime (PA) ligand. From the pattern, it can be observed that the characteristic sharp peak of crystalline polyacrylonitrile (PAN) found at θ = 17.1° was vanished, indicating the successful conversion of nitrile groups of PAN to amidoxime groups, ultimately forming the amidoxime chelating ligand [34]. Figure 6d displayed the XRD pattern of amidoxime chelating ligand after complexation with Cu(II) ions. It can be observed that there were three new reflections at 2θ = 43.3°, 50.4° and 74.1° indexing with (111), (200), and (220) copper planes, confirming the formation of PA–Cu complex [32].

### 3.5. Thermogravimetry Analysis

Thermogravimetry analyses (TGA) were carried out on the final and intermediate products such as the poly(hydroxamic acid) and poly(amidoxime) ligand including the jute cellulose, PMA-grafted cellulose, PAN-grafted cellulose. TGA scans were performed by using heating rate 10 min^−1^ under N_2_ gas atmosphere (Mettlar Toledo TGA/DSC +3). Figure 7a show the TGA scan of the jute cellulose, the thermal stability of jute cellulose is low in the higher temperature range from 310–800 °C relative to other samples, except PMA-grafted cellulose. A noteworthy weight loss of about 81% at 370 °C was found for cellulose materials due to the degradation of the hydroxyl groups (OH) and CH_2_OH function groups were present in the cellulosic materials [35]. 

Two-stage weight loss of the PMA-grafted cellulose was evident at 210–450 °C as shown in Figure 7b. In the first stage, the degradation of the acrylate function groups might be losses about 16% at 375 °C. In the second stages higher degradation of the PMA alkyl chains was observed, the weight loss was 92% at 446 °C. The total weight loss was 95%, which causes the liberated volatile gases such as CO, CO_2_, CH_4_, NH_4_ and HCN. During second stage thermal degradation the PMA-grafted cellulosic copolymer shows noticeable weight loss due to the breakdown of OH and CH_2_OH function groups was present in the cellulosic materials [35,36,37,38].

TGA curve (Figure 7c) for PHA ligand showed the almost steady state degradation from the room temperature to 800 °C. PHA curve crossed the jute cellulose curve at 375 °C where weight loss of cellulose and PHA are 46%, and then PHA ligand is stable than cellulose and PMA-grafted cellulose until 800 °C (82% loss). However, slight two-stage degradation was found when the weight loss started at 30 °C and continued to 105 ℃ (11% loss) due to the degradation of the water and the second stage loss up to 420 °C (60%). poly(hydroxamic acid) ligand curve exhibit two cross-section points, at 350 °C to cellulose and at 425 ℃ to PMA-grafted cellulose. Beyond the second cross-section point, the polymer ligand was more stable than cellulose and PMA-grafted cellulose until 800 ℃ (84% loss). In this PHA–copper complex was more stable than all others product as expected, however small degradation at 150 °C with 15% weight loss and 60% weight loss were found at the end 800 °C (Figure 7d).

The poly(acrylonitrile)-grafted cellulose (Figure 7e) showed one stage weight loss started at 290 ℃ with a small weight loss about 3%. After this temperature the degradation enhanced up to 480 ℃ with 52% weight loss causes the degradation of the functional groups (CN). In this stage, the volatile gases such as CO, CO_2_, CH_4_, NH_4_, and HCN can be liberated to enhance the degradation event at higher temperature [36,37,38].

The poly(amidoxime) ligand displayed two stages of degradation as shown in Figure 7f. In the first stage, a temperature between 80 and 200 °C (10% loss) was attributed to the slight loss of the NH_2_ functional groups. The second stage degradation occurs between 210 and 341 ℃ (40% loss). Poly(amidoxime) ligand exhibited two cross-section points, jute cellulose at 340 °C and PAN-grafted cellulose at 435 °C. Beyond the second cross section point, the polymer ligand showed higher stability compare to the jute cellulose, but similar stability compared to the PAN-grafted cellulose until 725 °C. It was observed that high water content in the poly(hydroxamic acid) and poly(amidoxime) ligands with their copper complex confirms that the polymer ligands show a hydrophilic materials. The poly(amidoxime) ligand-copper complex have a cross-section with poly(amidoxime) ligand at 280 °C, cellulose at 330 ℃ and PAN-grafted cellulose at 385 °C, after this temperature copper complex is more stable than all others produced as expected and total weight loss was fund 58% at the end of 800 °C (Figure 7g).

### 3.6. Adsorption Study of Poly(Hydroxamic Acid) and Poly(Amidoxime) Ligands

#### 3.6.1. Effect of pH on the Adsorption of Metal Ions by PHA Ligand

To examine the effect of pH on the adsorption behavior of the PHA ligands, four metal ions (Cu^2+^, Co^2+^, Cr^3+^, Ni^2+^) were chosen for the batch adsorption experiment ranging from pH 3 to 6. During the batch adsorptions, ammonium acetate was used as a buffer to adjust to the desired pH (3, 4, 5 and 6) in the aqueous heavy metal solutions. 

Figure 8a shows the adsorption capacity (q_e_) of heavy metal ions increased gradually from pH 3 to 6. According to the results obtained, Cu^2+^ showed the high adsorption capacity 352 mg/g at pH 6, indicating the highest binding capacities when compared with other three different heavy metal ions (Co^2+^, Cr^3+^, and Ni^2+^). The binding capacities of Co^2+^, Cr^3+^, and Ni^2+^ were 318, 230 and 188 mg/g, respectively at pH 6. A moderate increase of heavy metal adsorption capacities when pH was increased from pH 3 to pH 6 and pH 6 was the highest adsorption capacity set among all four pH conditions. Based on the results, the uptake of heavy metal ions by the PHA ligand is pH dependent. The strength of adsorption capacity of heavy metal ions by PHA ligand was in the order of Cu^2+^ > Co^2+^> Cr^3+^ > Ni^2+^. The uptake of heavy metal ions by the PHA ligand caused the complexation by coordination of PHA ligand with heavy metals.

The presence of bidentate hydroxamate anions formed a complex with a five-membered ring, where two O atoms of the acidic functional groups bound to the heavy metal ions to produce the chelates [19]. The adsorption uptake of heavy metal ions was induced due to the presence of the hydroxamic acid chelating group-CH_2_(C=O)NH(OH) that was amphoteric in nature [39]. It is an established principle that a metal ion binding event takes place by deprotonation of the OH group on the hydroxamic acids chelator, and following (O, O) coordination happens between the metals, certainly transition metals and the oxygen (C=O) and deprotonated OH group [40]. Previous studies have been accomplished on the primary hydroxamic ligands and specified two (O, O) bonding modes were complex formation with the metal ions, e.g., Cu(II) and V(IV) from the initial deprotonation and incorporates the coordination of the (CO)NHO moiety [40].

#### 3.6.2. Effect of pH on the Adsorption of Metal Ions by Poly(Amidoxime) (PA) Ligand

A similar batch adsorption study was performed by PA ligand as shown Figure 8b. To examine the effect of pH on the adsorption behavior, four heavy metal ions (Cu^2+^, Co^2+^, Cr^3+^, Ni^2+^) were also chosen for the batch adsorption experiment ranging from pH 3 to 6. Based on the results obtained, Cu^2+^ showed the highest adsorption capacity of 310 mg/g at pH 6. The binding capacities of Co^2+^, Cr^3+^, and Ni^2+^ were 295, 227 and 175 mg/g, respectively, at pH 6. The uptake of heavy metal ions by the PA ligand was also pH dependent and also similar strength of adsorption capacity by PA ligand as compared to PHA ligand. The presence of bidentate amidoxime chelate formed five-membered ring complexes, where two nitrogen atoms bound to the heavy metal ions to produce the chelates [11]. The uptake of heavy metal ions by PA ligand was also induced due to the presence of the amidoxime –(C(NH_2_)=N-OH) functional group in the ligand that has amphoteric features [39]. The same event which occurred in the hydroxamic acid can be observed for amidoxime ligand resulted in at optimal pH conditions (pH 5–6). Thus, the chelating ability of basic amino group of poly(amidoxime) towards heavy metal ions increased. [40].

### 3.7. Adsorption Kinetic Studies

The adsorption contact time required by the synthesized PHA/PA ligands to complex with selected heavy metal ions (Cu^2+^, Co^2+^, Cr^3+^, and Ni^2+^) from aqueous solutions were determined using batch adsorption method. The adsorption capacity for each time interval (0, 5, 15, 30, 60 and 120 min) was determined using ICP-OES. All the heavy metal ions showed the similar trend of adsorption capacities by PHA/PA ligands as a function of time, i.e., the adsorption capacities were increased from 0 min to 120 min (saturation point).

In the early time of adsorption, a great number of the amphoteric functional groups of PHA/PA chelating ligands are available to be bound with heavy metal ions via complexation by means of electrostatic attraction. The adsorption rate gradually decreased as the time interval increased up to 120 min and eventually reached a plateau, indicating that the rate of adsorption after the time of 120 min was stagnant. This is because as the time interval increased, available adsorption sites were decreasing gradually as many of the adsorption binding sites of the PHA/PA ligand were used for the binding of the heavy metal ions at the beginning of the time interval, and thus, fewer adsorption sites were available for the binding of the heavy metal ions at the later time intervals [14].

The rate mechanism of ligands with metal ions exhibited a major complexation reaction compared to that of ion-interchange reaction and hydrogen connecting mechanism. The kinetic behavior on the metal ions by the synthesized PHA/PA ligands at different times was studied at pH 6. The kinetic order was then determined using the kinetic model to have a better understanding on adsorption behaviors such as the adsorption and adsorption rate, and also the adsorption proficiency of the PHA/PA ligands.

#### 3.7.1. Pseudo First-Order Rate of Adsorption

In order to measure the kinetic parameters of pseudo first-order rate of adsorption, the following Equation (3) was applied [16]:(3)$$\log(q_{e}-\;q_{t})=\;\log q_{e}-\;\left(\frac{K_{\mathrm{ads}}}{2.303}\right)t$$

According to Equation (3), q_e_ and q_t_ are the adsorbed metal ions (mg/g) at equilibrium time and interval time in minutes, respectively, while K_ads_ is the pseudo first-order reaction rate constant (min^−1^) of the adsorption. The value of the q_t_ and K_ads_ were measured from the graph of the first order by the location of intercept and slope of plot of log(q_e_-q_t_) against time as shown in Figure 9a for PHA ligand and Figure 9b for PA ligand. Table 1 displays the results from the data estimation for the adsorption of four heavy metal ions by PHA/PA ligands.

The parameters are from the plot of Figure 9 and results are presented in Table 1. It can be seen that the adsorption capacity was in the order of Cu^2+^ > Co^2+^ > Cr^3+^ > Ni^2+^. However, the R^2^ values of Cu, Co, Cr, and Ni for PHA ligand were all significant (R^2 >^ 0.98), which are 0.987, 0.984, 0.982 and 0.977, respectively. On the other hand, the R^2^ values of Cu, Co, Cr, and Ni for PA ligand were insignificant, 0.935, 0.948, 0.982 and 0.895, respectively. In addition, there were relatively small differences in the adsorption between the experimental adsorption capacities (q_e_) and the calculated adsorption capacity, q_t_ by PHA ligand (Table 1). Therefore, those results suggested that the pseudo first-order rate of adsorption of the kinetic model was a good fit to the experimental data by PHA ligand, which is in good accordance with previous work [16]. However, there were significant differences in the adsorption capacities between the experimental adsorption capacities (q_e_) and the calculated adsorption capacity, q_t_ by PA ligand due to a lesser fit to the pseudo first-order rate of adsorption (Table 1), similar behaviour observed in previous work [21].

#### 3.7.2. Pseudo Second-Order Rate of Reaction

In order to determine the kinetic parameters of pseudo second-order rate of adsorption, we use the following Equation (4) [41,42]:(4)$$\frac{t}{q_{t}}=\frac{1}{k_{2}q_{e}^{2}}+\frac{t}{q_{e}}$$
where k_2_ is the pseudo second-order rate constant (g/mg min), q_t_ is the metals adsorption (mg/g) at particular time interval and the q_e_ is the metal ions adsorbed (mg/g) at equilibrium. 

The value of k_2_ and q_e_ were calculated from the plot of t/q_t_ against time as shown in Figure 10a for PHA ligand and Figure 10b for PA ligand. Table 2 displays the data results from the calculation for the adsorption capacities of four selected metal ions by PHA/PA ligands. 

According to the calculated results obtained in Table 2, the adsorption capacity was in the same order as of first order. The R^2^ values of Cu, Co, Cr, and Ni were in acceptable ranges (R^2^ > 0.98) for the PHA ligand; however, the experimental values (q_m_) of adsorption capacities and their respective calculated q_t_ values are significant differences (Table 2). Therefore, the pseudo second-order rate of kinetic model was not a good fit with the experimental values obtained [16]. 

On the other hand, R^2^ values of Cu, Co, Cr, and Ni were also in acceptable ranges (R^2^ > 0.98) for the PA ligand. Thus, the pseudo second-order rate of kinetic was good fit to experimental values (q_e_). This indicated that the adsorption mechanism involved in the removal of heavy metal ions by the PA ligand was chemisorption [21]. This means that the adsorption mechanism was chemically rate controlling. It involved the valance forces of attraction such as sharing or exchanging of electrons between the polymeric PHA ligand and the heavy metal ions [25]. However, we can observe that two different ligands behave differently due to their ligands coordination is different as PHA contains –C=O and PA contains –NH_2_ functional groups which act as a chelating agent [14,15,16,21].

### 3.8. Sorption Isotherm

Isothermal studies were carried out in order to study the effect of the concentrations of heavy metal ions on the adsorption capacity by PHA/PA ligands. Isothermal studies were carried out with four selected metal ions (Cu^2+^, Co2^+^, Cr^3+^, and Ni^2+^) via a batch adsorption technique, where all the parameters were set to be constant except the concentration of the heavy metal ion solutions used in ranging from 10–1800 ppm, as shown in Figure 11a for PHA and Figure 11b for PA ligand. It can be seen that increasing initial concentration of metal ions resulted in increasing adsorption capacity by the PHA/PA ligands towards those heavy metal ions. The binding capacity by the PHA/PA ligands gradually increased to a certain limit where it eventually reached the equilibrium and showed a plateau on a graph.

In order to find the better understandings of the isothermal behaviors of the adsorption process, there are some common isothermal models commonly used in the studies of the effect of concentration of an adsorbate by an adsorbent, such as the Langmuir isotherm model and Freundlich isotherm model. In this research, both Langmuir and Freundlich adsorption isotherm models were used to study the isothermal behavior of the adsorption of heavy metal ions by PHA/PA ligands. Both models were derived in linear forms before determining all the isothermal parameters. Linear forms of isothermal models were used instead of non-linear models due to the mathematical simplicity of the linear form model.

#### 3.8.1. Linear Langmuir Adsorption Isotherm

According to the assumptions of the Langmuir adsorption isotherm model, it is stated that an adsorption system that fits Langmuir model is that having four criteria: the adsorbent surface is uniform, where all the adsorption sites are equivalent; adsorbed molecules do not have interaction within each other; the adsorption system happens with same mechanism; and lastly, it involves one single layer or monolayer adsorption, which means the adsorption occurs only on the surface of the adsorbate [43]. The linear form of the Langmuir adsorption isotherm model is shown in the following Equation (5) [44]: (5)$$\frac{C_{e}}{q_{e}}=\frac{1}{q_{\max}K_{L}}+\frac{C_{e}}{q_{\max}}$$

As referred to the Equation (5), q_e_ stand for the equilibrium adsorption capacity (mg/g), C_e_ is the equilibrium concentration of heavy metal ions in their respective aqueous solutions (mg/L), q_max_ and K_L_ stands for the maximum capacity (mg/g) and Langmuir adsorption constant (L/g), respectively.

By applying Equation (5), all the values of q_max_ for heavy metal ions were determined from the slopes or gradients of the line graphs, while all the values of K_L_ for each heavy metal ion were calculated based on the intercepts obtained from the line graphs in the linear plots of C_e_/q_e_ against C_e_ as shown in Figure 12a for PHA ligand and Figure 12b for PA ligand.

The data presented in Table 3 show that all the values of correlation coefficient (R^2^) were all significant, R^2^ > 0.99. According to the calculated results recorded in Table 3, the adsorption strength of the metal ions are of Cu^2+^ > Co^2+^ > Cr^3+^ > Ni^2+^, which were 299, 257, 195 and 144 mg/g, respectively. Therefore, all the adsorption capacity values calculated from the Langmuir model have significant difference with the experimental adsorption capacities, q_e_ (Table 3). Although the values of R^2^ > 0.98 are an acceptable correlation, q_max_ data was not close to q_e_, thus the single or monolayer adsorption of heavy metal ions seems not to have occurred on the surface of the PHA ligand. Same is true for PA ligand, there q_max_ and q_e_ vales show the significant differences although R^2^ > 0.98.

#### 3.8.2. Linear Freundlich Adsorption Isotherm

According to the assumptions of the Freundlich adsorption isotherm, it is stated that an adsorption system having Freundlich isotherm model is multiple layer adsorption, whereby the surface of the adsorbent (polymeric ligand) is heterogenous [45]. A linear Freundlich isotherm is expressed by Equation (6) [44].
(6)$$\log q_{e}=\log K_{F}+\frac{\log C_{e}}{n}$$

Based on Equation (6), q_e_ stand for equilibrium adsorption capacity on the PHA/PA ligands, C_e_ is the equilibrium concentration (mg/L), K_F_ is Freundlich’s constant (L/mg), and 1/n is the heterogeneity factor that relates to the adsorption capacity.

By applying Equation (6), the values of n for every heavy metal ion were determined from the slopes or gradients of the line graphs, while all the values of K_F_ for each heavy metal ion were calculated based on the intercepts obtained from the line graphs in the linear plots of log q_e_ against log C_e_ as shown in Figure 13a for the PHA ligand and Figure 13b for the PA ligand. From the data obtained in Table 4, it can be observed that overall the values of correlation coefficient were significant, where R^2^ < 0.99. By making a comparison with the results in Langmuir adsorption isotherm, it can be observed that the correlation coefficient R^2^ values in the Freundlich isotherm fit better compared to the Langmuir isotherm, in which the R^2^ values of all heavy metal ions in Freundlich isotherm were more than 0.99, indicating Freundlich adsorption was more predominant compared to Langmuir isotherm. Thus, this indicated that it was likely to have multilayer adsorption of heavy metal ions occurring on the heterogeneous surface of the PHA/PA ligands [45].

### 3.9. Investigation of Adsorption Mechanism by X-ray Photoelectron Spectroscopy (XPS) Analysis

Amidoxime and hydroxamic acid are the bidentate ligands, both are strongly bound with copper ions owing for high chelating affinity towards the metal ions. The X-ray photoelectron spectra (XPS, PHI VersaProbe II) were obtained to explain the binding mechanisms of Cu(II) on the both PA and PHA ligands. The wide-scan XPS spectra of PA and PHA ligand binding with copper are shown in Figure 14a,b, respectively. In the case of the PA ligand with copper, binding energy (BE) peaks were found at 287.5, 403.0 and 5334.6 eV corresponding to the C1s, N1s, and O1s spectra, respectively. The binding of Cu(II) exhibit two peaks with BEs of 936.5 and 957.0 eV for the signals of Cu2p3/2 and Cu2p1/2 (Figure 14a). Similar spectra were obtained for the case PHA ligand binding with copper, as such BEs observed at 288.1, 402.6 and 535.2 eV corresponding to the C1s, N1s, and O1s spectra, respectively. Also Cu(II) bind with ligand showed two peaks with BEs of 937.0 and 957.0 eV for the signals of Cu2p3/2 and Cu2p1/2 (Figure 14b).

The N 1s spectra of the core-level using PA ligand (Figure 14a) displayed two peaks with BEs of 399.3 and 400.6 eV, representing to N atoms in C=N–OH and NH_2_–Cu species, respectively. Later peak at 400.6 eV of the N 1s was accountable for the coordinated bond of the amide with copper. In principle, the coordination mechanism of the lone pair of electrons in the N atoms was contributing to the Cu(II) ions emerging in the reduction of the electron density of the N atom to attain increased BE [16]. The XPS is a high-powered method to explore the electron exchange between the donor and acceptor species.

On the other hand, the oxygen atoms in the CO species promote the coordinate bonds between PHA ligand and copper ions (Figure 14b). In this case, the O 1s core-level spectrum appeared an extra peak at 535.7 eV (Figure 14b) due to the O atoms are electron donors for the coordination bond to copper ions [15]. The lone pair of electrons in the oxygen atoms of the PHA ligand donated the electron pair to form a coordination bond between Cu(II) and the O atoms. As a result, the electron cloud density of the oxygen atom was reduced to attain higher BE [40]. Thus, it can be resolved the binding of copper ions on the PHA ligand is related to oxygen atoms belonging to the hydroxamate groups. 

### 3.10. Reusability Studies of Poly(Hydroxamic Acid)/Poly(Amidoxime) Ligands

It is important to ensure reusability in real-life applications, such as in the removal of industrial effluents in factories to prove its cost-effectiveness. A reusability study of PHA/PA ligands involved the process of elution and regeneration of the PHA/PA ligands in adsorption-desorption properties. The adsorbed heavy metal cations can be extracted or desorbed from the adsorbents under very low pH condition [14,15,21]. Thus, by applying the 2M of HCl solution was used to extract the adsorbed heavy metal cations from the adsorbents.

In this study, PHA/PA ligands (150 mg) were used to adsorb the copper ions and metal ions were desorbed by using 20 mL of 2M HCl, and after washing and drying, this was reused again for a new adsorption process. Seven cycles of adsorption-desorption processes were performed using the same metal complexes. The percentage of copper sorption was determined for each cycle. In Figure 15a, it can be observed that the efficiencies of sorption 93% during extraction was 92% after 7 cycles of adsorption-desorption processes for PHA ligand. On the other hand the efficiencies of sorption was found to be 93% during extraction reaching 91.5% after 7 cycles of adsorption-desorption processes for PA ligand as shown in Figure 15b. The reusability experiment indicated that there was no significant loss of the adsorption capacity of copper ions for the reused PHA/PA ligands. Therefore, it can be concluded that both the ligands may suitable to be used in the application of extraction of heavy metal ions in the industrial wastewaters.

### 3.11. Electroplating Wastewater Purification

#### 3.11.1. Practical Application of PHA Ligand

The effects of parameters such as pH, contact time, concentration of adsorbate and elution of ligand was tested in the previous section to determine the effectiveness of using PHA ligand in the removal of toxic heavy metal ions not only in lab scale study but also in real-life practical applications. Table 5 shows the results obtained for the concentrations (before ligand treatment and after ligand treatment) by using ICP-MS. Several transition metal ions including alkali and alkaline earth metal ions were found in industrial wastewater samples (Sample 1 and Sample 2) collected from a semiconductor metal plating workshop that performs printed circuit board (PCB) etching (Porcel, Singapore).

In Table 5, the results obtained showed that the concentrations of metal ions in both the wastewater samples decreased after being treated with synthesized PHA ligand under optimal conditions. According to the results, it can be seen that the removal of transition heavy metal ions were relatively effective and significant, such as Fe^3+^, Cu^2+^, Pb^2+^, Zn^2+^, Cr^3+^ and Ni^2+^ which are common toxic heavy metal ions that can be found in most industrial effluents.

In the case of Sample 1, the removal efficiency of heavy metal ions Cu^2+^ was even higher than the other heavy metal ions, in which of them showed the removal efficiency by PHA ligand of more than 98%, while removal of Fe^3+^, Pb^2+^, Cr^3+^, Mn^2+^ and Ni^2^ was also more than 90%. In case of Sample 2, the removal efficiency of heavy metal ions of Fe^3+^ and Cu^2+^ was even higher at more than 98%. Other transition metal ions also show removal efficiency more than 90%. Overall, the highly effective removal of heavy metal ions by PHA ligand was found. Other metal ions were also detected in both the industrial wastewater samples, such as Ba^2+^, Ca^2+^, Mg^2+^, K^+^, Rb^+^, Na^+^, Ag^+^, V^4+^ and Al^3+^, and they have shown lower removal efficacies due to the nature of hydroxamic acid chelating ligand who does not selective for alkali metals.

#### 3.11.2. Practical Application of PA Ligand

The PA ligand has been used for the removal of toxic heavy metal ions in real-life practical applications. Table 5 shows the analyses using ICP-MS on the concentrations (before and after ligand treatment) of several types of transition heavy metal ions presented in two industrial wastewater samples (Sample 1 and Sample 2) collected from a semiconductor electroplating factory undertaking PCB etching (Porcel in Singapore).

According to the results obtained (Table 6), it can be determined that the removal of transition heavy metal ions were relatively effective and significant, such as Fe^3+^, Cu^2+^, Pb^2+^, Zn^2+^, Cr^3+^ and Ni^2+^ which are common toxic heavy metal ions that are abundantly found in most industrial effluents or wastewaters. The removal efficiency of heavy metal ions such as Fe^3+^ and Cu^2+^ was even higher than the other heavy metal ions, in which 98% of both of them in the water samples were removed by the PA ligand. Removal of other transition metal ions such as Pb^2+^, Zn^2+^, Cr^3+^ and Ni^2+^ also had relatively high removal efficiency, where overall removal were more than 90%. Alkali earth metals such as Ba^2+^, Ca^2+^, Mg^2+^, alkali metals such as K^+^, Rb^+^, Na^+^, and some other metal ions such as Ag^+^, V^4+^ and Al^3+^, the removal was relatively lower due to the nature of the amidoxime ligand. It can be concluded that the synthesized PA ligand is very useful and effective especially in removing toxic transition heavy metal ions that are commonly found in industrial wastewater. 

### 3.12. Comparison with Other Adsorbents

The pollution of surface and groundwater with heavy metals is a vital concern globally. Overabundance of these elements poses severe health risks for humans and other life forms through bioaccumulation along food chains. Therefore, steps have been taken to reduce the amount of such elements in water to acceptable levels [46]. To reduce the toxicity, many scientists have anticipated that low cost and biodegradable polymers such as cellulose can be modified with high potential to evacuate the metals ions to solving these environmental problems. Based on the current research, functional groups such as carboxyl, amino, amine group or sulfur group in the polymer potentially focus on the binding of metal ions on the cellulose-based adsorbents and also affect the binding ability and the stability of the complex ligand [47,48]. The acrylamide monomer was grafted on the surface of cellulose and the resulting bio-ligand has COO-, -NH_2_, and -OH functional groups and performed well in the extraction of toxic metals [49].

Several studies showed that chemical modification is the best and easiest way to modify the cellulose surface via directly attaching or grafting the selected monomer on the primary or secondary hydroxyl group [14,15]. The hydroxyl groups on the cellulose surface were implemented to functionalize with selected monomer or functional group can successively form the known chelating ligands. The monomer was grafted onto the cellulose as the side chains covalently attached to the main chain of the cellulose backbone through ionic (free-radical initiating processes) [15]. To create highly reactive radical in the various sites of cellulose using high-frequency ultrasound [50], ceric ammonia nitrate [11,21], cobalt(III) acetylacetonate complex salts [51], and free radical generators such as azobisisobutyronitrile [52] were used with a profound effect on the initiator due to it being able to provide high efficiency in the grafting process.

Table 7 shows the different type of chemically modified grafted cellulose with various modifying agents/ligands groups, adsorbates and their adsorption capacities for different adsorbents [11,14,15,21,53,54,55,56,57,58]. The lone pair of electrons present on the nitrogen atom of amidoxime and oxygen from hydroxamic acid induced a coordination bond with the metal ions. Thus, amidoxime and hydroxamic acid groups have a bidentate character which donates a proton and a basic lone pair of electrons on the nitrogen and oxygen to coordinate with the metal ions [9]. However, various adsorbents were prepared from the cellulose substrate, which successfully was utilized to remove or extract heavy metal ions from surface or industrial wastewater as shown in Table 7.

## 4. Conclusions

Cellulosic materials can be obtained from various natural sources and can be employed as cheap adsorbents. Furthermore, chemical modification of the cellulose materials is necessary for better adsorption capacities enabling heavy metal recovery from wastewater. Both poly(hydroxamic acid) and polyamidoxime chelating ligands were synthesized by utilizing cellulose extracted from Jute fiber. The synthesized PHA and PA ligands were characterized by a batch adsorption study of copper, chromium, cobalt and nickel ions using the effects of pH, sorption contact time (kinetic), isotherm, and practical applications of synthesized ligands were determined. The Freundlich isotherm showed the best fit to the linear isotherm model with excellent correlation coefficients for all heavy metal ions, indicating that multiple layer adsorption happened on the PHA/PA adsorbents. Both polymeric ligands exhibit outstanding characters for heavy metal removal from industrial wastewater, with up to 98% removal of copper and other metal ions up to 90%, from electroplating wastewater.

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
