# Peer review of "Polymer Ligands Derived from Jute Fiber for Heavy Metal Removal from Electroplating Wastewater"

_polymers, 2020, doi:10.3390/polym12112521_

Round 1

Reviewer 1 Report

Poly(hydroxamic acid) and poly(amidoxime) ligands were prepared from the PMA-grafted cellulose and PAN-grafted cellulose in the manuscript. The polymeric ligands showed outstanding heavy metals removal capacity from the industrial wastewater. The work is very interesting and I recommend it publishing in Polymers.

1. Equation 1 and equation 2: "L" should be changed to "m".  Please kindly see the following reference.

Xiaoyan Cao, Qing Wang, Shuai Wang, et al. Preparation of a Novel Polystyrene-Poly(hydroxamic Acid) Copolymer and Its Adsorption Properties for Rare Earth Metal Ions. Polymers, 2020, 12, 1905, doi:10.3390/polym12091905

2. Scheme 1: "+Mn" should be changed to "Mn+".

3. Figure 3 and Figure 4:  The texts in figure (a) are different with the texts in other figures. The width and height of the scale bars in figure (a) are also different with the others.

4. Figure 7: The annotations in the figures are inconsistent with the description in the figure title. For example, in the left fagure, the line b (blue line) is poly(hydroxamic acid), but in the title of Figure 7, b is PMA-grafted cellulose.

Author Response

Please see attached PDF file for reply to the reviewer comments.

Reviewer 2 Report

The manuscript reports modification of cellulose obtained from jute lignin using free radical grafting reaction to obtained materials which are afterwards used for heave metal ion adsorption. The manuscript is interesting, however I think some issues must be addressed before it can be published. Below I made a list of my comments:

  • Line 16: “Adsorption study with some heavy metal ions” => it should be mentioned which ones in the abstract.
  • Line 38: “However, it will be a very big trouble” => I think that the language is to informal, please check the manuscript for that.
  • Line 39 (and some others): I think that there is a double space, please search whole document for that mistake.
  • Line 47 – 48: more citations should be placed there.
  • Scheme 1: “Cell” is a word in English. Please use some other abbreviation (like CL or something similar) which is no longer a regular word.
  • What I am missing is adsorption data etc. of pure cellulose without modification (as a reference). If it is available in literature I would cite it (if the Authors don’t have the corresponding data).
  • The reusability experiments should be added if the material is planned to be used in water purification.
  • A comparison with other adsorbents would improve the manuscript.
  • Freundlich constant: the unit is different than the one given in a table. Please check in the literature, as I recall there is even a paper about how to obtain correct unit for this model.

Author Response

(The authors gave the same response as above.)

Round 2

Reviewer 2 Report

The Authors have nicely addressed all the comments.

Maybe one thing should be corrected, the Cell was changed into CL only in one place, whereas in Scheme 1 it still has not been corrected, however, this can be changed during the proof correction.